# Does Bedtime Really Matter? Examining How Sleep Timing Relates to Sleep Duration and Overweight Status in Midwestern Latine Youth

**DOI:** 10.3390/children13010032

**Published:** 2025-12-26

**Authors:** Blake L. Jones, Bethany Lundy, Dakin Stovall, Benjamin D. Seely, Kelsey Zaugg, Joshua Castro, Kara M. Duraccio, Chad D. Jensen, Tanya Austin, Zoe E. Taylor

**Affiliations:** 1Department of Psychology, Brigham Young University, Provo, UT 84602, USA; lundy8@student.byu.edu (B.L.); dakinstovall@gmail.com (D.S.); bdseely@byu.edu (B.D.S.); kelsey.zaugg@hsc.utah.edu (K.Z.); cas18056@student.byu.edu (J.C.); kara_duraccio@byu.edu (K.M.D.); chad_jensen@byu.edu (C.D.J.); tmata121@student.byu.edu (T.A.); 2Department of Human Development and Family Science, Purdue University, West Lafayette, IN 47907, USA; zetaylor@purdue.edu

**Keywords:** sleep duration, bedtime, wake time, overweight status, Latine youth

## Abstract

**Highlights:**

Increased sleep duration, along with earlier bedtimes, seem to promote health by decreasing overweight risk in Midwestern Latine youth. Earlier bedtimes are associated with lowering the risk for being overweight, even when controlling for sleep duration.

**What are the main findings?**
Shortened sleep duration is related to increased risk of being overweight.Later bedtimes are related to risk of being overweight, even when factoring in sleep duration.

**What are the implications of the main findings?**
Interventions for preventing or intervening with overweight risk should consider sleep as a potentially important factor in determining health.Traditional studies have focused on sleep duration but sleep timing may play a unique and important role in decreasing overweight risk.

**Abstract:**

**Background/Objectives**: Overweight and obesity is a continuing health concern for preadolescent youth. We assessed associations between sleep timing and sleep duration and body mass index/body composition in Latine youth. **Methods**: Participants were 119 Latine youth (mean age 11.53 year; 58.8% girls) and their mothers living in the rural Midwestern U.S. Youth reported their average bedtime and waking time. Heights and weights for children and mothers were measured by trained research assistants and were used to calculate BMI scores (in mothers), as well as BMI percentiles and overweight status (in youth). Mothers completed surveys for demographic variables. **Results**: Youth who went to bed before 9:30 PM (mean bedtime) obtained more sleep than those with later bedtimes (9.73 h vs. 8.63 h, respectively, *t*(117) = 7.88, *p* < 0.001). Each extra hour of sleep duration was associated with a decreased risk of being overweight (*OR* = 0.53 for weeknight sleep, *OR* = 0.67 for weekend night sleep), and each hour later to bed was related to increased risk for being overweight (*OR* = 2.35 on weeknights, and *OR* = 1.66 on weekend nights). To replicate previous work, we broke the youth up into four sleep timing groups: early-to-bed and early-to-rise (EE), early-to-bed and late-to-rise (EL), late-to-bed and early-to-rise (LE), and late-to-bed and late-to-rise (LL). Youth with LL sleep patterns on weeknights were much more likely to be overweight compared to youth with EE patterns (*OR* = 4.94). On weekend nights, compared to EE weekend youth, LE and LL weekend youth were more likely to be overweight (*OR* = 3.45 and *OR* = 3.32, respectively). Wake times were not significantly related to overweight risk. **Conclusions**: Sleep timing patterns, especially sleep duration and earlier bedtimes, may be important to address in future research on obesity interventions. Findings suggest that earlier bedtimes may play an important and complimentary role in health, in addition to sleep duration alone, and this study highlights the need for more research in underserved, minoritized populations.

## 1. Introduction

Childhood obesity continues to be an epidemic in the U.S. and presents a substantial public health concern [1,2,3,4]. Since the 1980s, obesity rates in children have nearly tripled [1]. Recent data from the National Health and Nutrition Examination Survey estimated about 16.1% of U.S. children and adolescents aged 2–19 were overweight, 19.3% were obese, and 6.1% were severely obese, with significant increases in both obesity and severe obesity observed particularly in Latine adolescents from the years 1999 to 2018 [2]. In obesity studies with adults, many researchers have used body mass index (BMI) scores to describe overweight or obesity cut-offs. However, when measuring overweight or obesity in pediatric populations (ages up to 19), the basic BMI score is not as meaningful and does not clearly identify weight status. In studies with children and youth, researchers and clinicians must measure the height and weight and then compare it to age-specific and sex-specific percentile growth charts that are calculated by groups such as the Centers for Disease Control (CDC) and other agencies. We then use these BMI percentiles, factoring in the growth chart calculations, to determine the cut-off for overweight status. In children and youth, overweight is defined as having a BMI at or above the 85th percentile [4].

Being overweight or obese can impact physical and psychological health and is associated with increased risk for health conditions such as asthma, cardiovascular disease, type 2 diabetes, high blood pressure/hypertension, non-alcoholic fatty liver disease, obstructive sleep apnea, and psychosocial difficulties [3]. Research on lifetime financial costs of obesity has indicated a proportional relationship between BMI and total cost, as well as more costs accounted for by lost productivity than by healthcare expenses alone [5,6].

Past research has established links between sleep and the risk for childhood obesity [7,8,9]. Increasing evidence supports associations between insufficient sleep duration and increased body mass index (BMI) across childhood and into adulthood [9,10,11,12,13]. Chen and colleagues [14] reported that children with shorter sleep duration experienced a 58% higher obesity risk, and children who had the shortest sleep duration had a 92% higher risk of obesity. Moreover, they found that, for each hour increase in sleep, the risk for overweight and obesity was reduced by 9% [14]. Although sleep duration on its own has been shown to be an important factor associated with obesity, other sleep factors and patterns, such as bedtimes and sleep quality, have also proven influential [15,16,17,18]. In a study by Olds and colleagues [19], they found that going to bed late and waking up late were associated with less physical activity and higher BMI in adolescents [19]. They also found that adolescents who went to bed late and woke up late were 1.5 times more likely to be overweight or obese than their early-to-bed and early-to-rise counterparts. Ultimately, they found that sleep timing was associated with activity patterns and weight status independent of sleep duration [19]. A major emphasis of the current study was to replicate similar work performed by Olds et al. (2011) [19] that late bedtimes would be related to overweight status even when controlling for sleep duration. More work in this area is needed because other studies have not reported similar findings to Olds et al. For example, Sunwoo and colleagues recently reported that sleep duration significantly predicted obesity in youth but sleep timing was not significant [9]. Therefore, more research is needed on different populations and across different developmental ages to gain a clear understanding of the differences from the relative contributions of these sleep factors.

Several studies have suggested additional mechanisms that might help explain associations between sleep duration, sleep timing, and obesity [17,18]. Notably, shorter sleep duration and later bedtimes have been shown to lead to higher energy consumption and lower energy expenditure [17]; increased obesogenic eating behaviors and shifted eating opportunity [17,18,20]; and changes in levels of the hormones ghrelin and leptin (both important for appetite control and energy expenditure) [21]. However, findings in children have had mixed results on the link between sleep and alterations in these hormones [22], suggesting that changes brought about through sleep may be dependent upon other factors, such as age and sex.

Generally, Latine adolescents are at increased risk for overweight/obesity, which may be contributed to by disparities in language, acculturation, access to health care/insurance, parental workload, socioeconomic status, among other social determinants of health [23,24,25,26,27,28,29,30]. However, little research has been performed in the past pertaining to rural Latine youth on these topics, and that research is even more sparse when it comes to Latine youth living in the rural Midwestern U.S. areas [31]. Most research on Latine youth in the U.S. is performed in the Southwestern US (e.g., Texas, California, Arizona), on the east coast (e.g., Florida, New York, or in metropolitan areas [25,26,27,28,29]. There is a lack of research regarding this population and the effects of stress, sleep, and dispositional features on overweight status, and how the factors are related [30,31], which is why the research was conducted. Researchers highlight the need to address multiple areas of sleep, beyond sleep duration, to provide a better picture of sleep habits and sleep disparities in Latine populations [30]. To address these gaps in the current literature, there is a need to explore the associations between sleep behaviors, both timing and duration, with obesity risk in rural Midwestern Latine youth.

From a theoretical perspective, this gap falls in place with two major supporting theories. First, Bronfenbrenner’s Ecological Systems Theory (EST) focuses on how development occurs within individuals that are nested within other systems such as biological factors, families, communities and culture [32]. Additionally, The Six Cs model extends the EST to focus on how overweight in childhood is related to competing spheres of influence: from the cellular level (biology and genetics) to the child level (individual behaviors) to the clan level (family and parent factors) to the community level (peers, school, and beyond locally) to the country level (national factors such as politics and economics) to the cultural level (societal norms and customs) [33]. In relation to the current study, sleep is just one of the many factors that are part of the child sphere, though it can be influenced by factors from the other spheres (e.g., parental sleep rules, peer norms, and cultural norms) [32].

The current study focused on sleep and obesity in the understudied population of Latine immigrant youth living in the rural Midwest. We examined these questions in relation to three separate hypotheses. First, we hypothesized that short sleep duration would be associated with increased risk of overweight in adolescents. Second, we hypothesized that later bedtimes would be associated with higher risk for overweight status, even when controlling for sleep duration. Finally, we hypothesized that our study results would replicate the findings from the approach used by Olds et al. [19], which compared four groups of sleep schedules (e.g., early-to-bed and early-to-rise (EE), early-to bed and late-to-rise (EL), late-to-bed and early-to-rise (LE), and late-to-bed and late-to-rise (LL)) and found that they had different prevalence levels of being overweight.

## 2. Materials and Methods

### 2.1. Participants

Data for the current study (*N* = 119) come from (Project SALUD) a cross-sectional observational study designed to assess health and well-being in Midwestern U.S. Latine youth and their parents. Inclusion criteria required both the child and biological mother to participate and each self-identify as being Latine and required the child to be in fifth or sixth grade (ages 10–12 years old). There were no specific exclusion criteria used in this study. The sample size for the study was based on an a priori power analysis using G*Power3.1, which helped us determine that we needed a sample size of at least 87 participants by using a median effect size, 80% power, and a two-tailed alpha of 0.05. With this number in mind, we set our goal to recruit 120 families to oversample and allow for possible missing data, to strengthen our power to potential significant associations if they were present, and to work within the financial resources we had available to do this field research and pay these families to participate.

Families were recruited using convenience sampling and by actively recruiting through the use of community resources (e.g., public schools, community events, and extension offices) after receiving institutional review board approval from (Purdue University). We worked with school administrators, to receive permission to send out pamphlets and fliers to Latine youth in fifth and sixth grades. We also worked with extension agents in county offices to have them help us actively recruit at community events such as parks, soccer games, libraries, grocery stores and farmers markets, and other social and school events. These extension workers were vital resources because they had previously built strong connections with this community and had already established trust and rapport with the families. In addition to these recruiting methods, we also used snowball/referral sampling to reach out to participants’ friends and neighbors who met the study criteria. This exhaustive effort was required to get our sample as close to 120 families as possible.

The mean age of the youth was 11.53 (*SD* = 0.69). A majority of the youth were female, *n* = 70 (58.8%), and a majority of the youth *n* = 101 (86.0%) were living in a two-adult household (i.e., biological parents were married, or their mother was living with a partner). The families reporting their annual household income, which averaged between USD 25,001 and USD 30,000. The majority of parents (69% of mothers, 79.2% of fathers) reported that they had not completed high school. Most of the families (78%) lived in smaller rural towns, with populations less than 20,000 people. The rest of the families (22%) lived in a midsized city of about 70,000 people.

### 2.2. Procedure

Interested participants reached out to us via phone or email or provided their contact information at public events if we found them through active recruiting at events. After a brief screening process over the phone, to make sure that they met the basic inclusion criteria, research assistants scheduled times with families to visit their homes. Eligible families were visited twice in their homes by trained bilingual research assistants (RAs). The RAs were required to be fluent in English and Spanish so that they could clearly communicate with the parents and youth. Prior to conducting any home visits, the RAs were trained on the procedures repeatedly to ensure that they could conduct the informed consent process, collect measurements following standardized protocols, and perform all other protocols consistently and within our training guidelines. They were also provided with lab phones so that they could contact the primary investigators in the event of any emergencies or questions that came up during the data collection visits (primarily because the participants lived up to an hour away from the university and we needed to be able to have a way to communicate with our RAs when it was necessary).

During the first visit, RAs explained the informed consent process. Then parents signed the written informed consent documents and youth signed the written child assent documents. Next, RAs measured heights and weights and gave participants copies of paper surveys (in either Spanish or English as preferred by the participant) to complete. After giving them the surveys and answering any questions the youth or parents had, the RAs left the home after the first visit. The vast majority of the youth completed surveys in English (98.3%), while the majority of mothers (92.4%) chose to complete surveys in Spanish. We instructed participants to complete the surveys privately, and then to seal them in provided manila envelopes to ensure privacy. It took up to 90 min for mothers, and 60 min for youth, to complete their surveys. The RAs then returned to participants’ homes approximately three days later for the second visit. In this visit, RAs collected the surveys (in the manila envelopes) and checked the surveys to ensure that they were completed. Then they paid the participants for their time. Mothers received $50 and youth were paid $40.

### 2.3. Measures

***Youth Body Mass Index (BMI) Percentile and Overweight Status***. Youth’s body mass index (BMI) was measured by two trained RAs using standardized CDC protocols to measure height and standing weight. The two trained RAs measured heights and weights in the homes using a SECA 213 portable stadiometer and a SECA 869 calibrated digital flat scale (with a cable remote display to provide discretion and comfort for participants). The RAs took the measurements independently and then compared measurements to one another. If the measurements were close, then the two measurements were averaged. If the height was off by more than 0.5 cm or if the weight was off by more than 0.1 kg, then one of the RAs took a third measurement of either the height or weight. We then took the average of the measurements and calculated the BMI percentile using the CDC Tools for Schools (CDC), which also accounts for child age and sex by using adjusted growth charts. After calculating the BMI percentiles, we also created a dichotomous ‘overweight status’ variable by breaking participants into two groups. The overweight group had a BMI percentile of 85.0 or higher, and the healthy weight group had a BMI percentile under the 85th percentile.

***Sleep Variables***. Sleep timing (bedtime and time waking up) were reported by youth self-report. For bedtime and wake time, youth were asked, “On school nights, what time do you usually go to bed?” and “On school days, what time do you usually wake up?” Next, youth reported their bedtimes and waking times on weekend nights using similar questions but starting with, “On weekend nights…” and “On weekend days…”, respectively. We then calculated average sleep duration on weekday nights and on weekend nights using bedtime and wake times. For this analysis, weeknight and weekend night sleep durations were analyzed as independent predictors of dependent variables.

***Sleep Timing Groups***. We created four distinct sleep timing categories by finding the median data points for bedtimes on weeknights (based on similar group categorizations used in Olds et al., 2011 [19]). We used median splits to try to obtain groups that were the most similar in size so that the comparisons in the analyses would be more equal. Youth were then dichotomized into “early-to-bed” if they were below the median, or into “late-to-bed” groups if they were at or above the median. Similarly, we found the median data point for wake time on weekdays. Following the same procedure, youth were categorized into “early-to-rise” or “late-to-rise” groups if they were below the median or if they were at or above the median, respectively. For each of these categories, the youth were assigned a 0 (early) or a 1 (late) for bedtime and wake time. We then computed four sleep timing categories based on these scores and assigned youth to early-to-bed/early-to-rise (EE); early-to-bed/late-to-rise (EL); late-to-bed/early-to-rise (LE); or late-to-bed/late-to-rise (LL). This process was repeated for weekend night sleep groups.

### 2.4. Statistical Analysis

We first examined the variables to assess normality in the data and look for any missing data. The data did not have any excessive skewness or kurtosis in the variables. There was no missing data for the self-reported sleep variables or the BMI measurements that our RAs collected (for mothers or youth), as well as for most of the control variables. The only missing data were for maternal education level (two missing), and family income (11 missing). We found that these were missing at random, so we used imputation to assign the variable mean to these values before using them in our correlational preliminary analysis, or as control variables in the analyses. Once we had the completed data available, we conducted preliminary analyses to assess correlations and descriptive statistics (means, medians, standard deviations, and range).

Because sleep duration is constructed using typical bedtime and wake time, we expected that these variables would be strongly correlated. To ensure that multicollinearity was not a problem in our analyses, we tested the variation inflation factors (VIF) for sleep duration and sleep timing variables using the collinearity diagnostics function in linear regression analyses. Generally, multicollinearity is seen to be a concern if VIF scores are above 10, with even the more conservative views suggesting that VIF scores below 5 indicate that there are no significant concerns for multicollinearity in the variables [34,35]. All of our scores were acceptable, with the highest score being 2.47, indicating that multicollinearity was not a concern with our sleep duration and sleep timing variables for both weekday nights and weekend nights. Additionally, we tested the assumptions for logistic regression using the Box–Tidwell tests and ensured that we had linearity of the logit for our continuous predictors.

Once we knew that the data met the assumptions for our analyses, we conducted logistic regression models to calculate odds ratios to assess risk for overweight. We first ran these models for continuous outcomes of sleep duration, and for the bedtime and wake time variables for weeknights and weekend nights. For the sleep timing group comparisons, we used the categorical function in the logistic regressions to compare the other sleep timing groups (EL, LE, and LL) to the EE timing group as the reference group. These models were run as direct, unadjusted models with only sleep variables as independent variables and the dichotomized risk for being overweight as the outcome variable. Finally, we ran adjusted models that controlled for youth age, sex, parent education level, family income level, and maternal BMI (which has been linked as one of the top predictors of child BMI [13]).

To help differentiate group differences between the sleep timing groups (bedtimes and wake times) we also ran and presented unadjusted ANOVA analyses to give more details about the groups. In these analyses, our intention was not to deviate from the hypotheses in this study, but rather to add clarity and present the basic differences across the groups for both weeknight and weekend night sleep and overweight prevalence data. These analyses highlight group differences in the specific sleep duration of each group, the average bedtimes and wake times of each group, and the percentage of youth who were overweight in each sleep timing group.

## 3. Results

*Preliminary Analyses*: We first ran correlational analyses and reported on the mean for the overweight status percentage, sleep variables, cortisol variables, and control variables (see Table 1). We found that the majority (60.5%) of youth were overweight. There was a significant difference by gender, with 79.6% of boys being overweight compared to 47.1% of girls being overweight, *t*(117) = 3.88, *p* < 0.001. Additionally, there was a significant difference in youth-reported sleep duration (based on our calculations from their bedtimes and wake times) on weekend nights, such that boys reported getting an average of 9.89 h of sleep on weekends and girls reported getting an average of 10.44 h of sleep, *t*(117) = −1.94, *p* = 0.028. Other than these two variables, there were no other significant differences between boys and girls in the study variables.

*Descriptive Sleep Statistics*. Average sleep duration on weeknights was 9 h and 12 min (*SD* = 56 min). On weekends, the average sleep duration increased to 10 h and 13 min (*SD* = 1 h, 32 min; see Table 1 and Table 2). Respective mean bedtimes during the weeknights and weekend nights were 9:36 PM (*SD* = 47 min) and 10:58 PM (*SD* = 1 h, 19 min). Mean wake times during the weekdays and weekends were 6:47 AM (*SD* = 36 min) and 9:11 AM (*SD* = 1 h, 15 min), respectively.

Hypothesis 1 was supported. We found that short sleep duration was associated with an increased risk of overweight/obesity in adolescents on weeknights and weekend nights, particularly when we adjusted for the control variables (race, ethnicity, age, sex, education level of the mother, family income, and maternal BMI). In these analyses, there was a decreased risk for being overweight for every additional hour of sleep the youth got on weekdays (*OR* = 0.53) and on weekend nights (*OR* = 0.67; see Table 3). These findings represented a 47% decrease in risk of being overweight for each additional hour of sleep during weeknights, and a 33% reduction in risk of being overweight for each additional hour of weekend night sleep.

Hypothesis 2 was also supported. Sleep timing was related to overweight status, such that the majority of youth with later bedtimes during the week (70.2%, *p* < 0.05) and on weekends (73.4%, *p* < 0.05) were overweight (see Table 3). Importantly, we found that later bedtimes were associated with higher risk for overweight status, even when controlling for sleep duration. Regardless of wake time, for each additional hour delay in bedtime during the week, adolescents’ risk for overweight increased (*OR* = 2.35; see Table 3). Similarly, each additional hour delay in bedtime on the weekends was related to a statistically significant increase in their risk for being overweight (*OR* = 1.66).

Hypothesis 3 was also supported. When we broke the group into four categories, similar to the Olds et al. (2011) [19] study, we found that the LL group had a significantly higher risk for being overweight when compared to the other groups (see Table 4. Compared to the reference group (EE), the LL group were 4.94 (*OR*) times more likely to be overweight than the EE group. The EL and LE groups were not significantly different from the EE group in our adjusted logistic regression analyses for weeknight sleep. This demonstrated that the late-to-bed group that also woke up later on weekdays was at the most risk for being overweight.

In our adjusted analyses for those group comparisons using weekend nights, both of the late-to-bed groups were significantly more likely to be overweight when compared to the EE reference group. The LE group was 3.45 (*OR*) times more likely to be overweight, and the LL group was 3.32 (*OR*) times more likely to be overweight. EL weekend night youth did not have a significant difference in risk of overweight compared to the EE weekend night youth (reference group). These analyses demonstrated that youth who went to bed later on weekend nights were more likely to be overweight, regardless of waking time (see Table 4).

## 4. Discussion

This study demonstrates that Midwestern Latine youth’s risk of being classified as overweight is significantly associated with sleep duration and sleep-timing patterns, supporting the hypotheses in our study. These results support previous studies that found associations between shorter sleep duration, sleep timing, and increased risk for childhood obesity/overweight found in previous studies [17,18,19,20,36,37,38]. Although these outcomes are supported by previous research [7,8,9,10,11,12,13,14,15], the results of this research suggested that sleep duration may not tell the entire story. In addition to adding to the literature on sleep duration predicting risk for overweight, the link between later bedtimes and overweight risk contributes to a smaller but growing number of studies looking beyond sleep duration as an important factor in health and well-being. Thus, when addressing sleep and health interventions in the future, it may be helpful to focus not only on the amount of sleep, but also on the timing when that sleep occurs [36,37,38].

In the current study, sleep duration and bedtimes were significantly associated with risk for overweight in both the continuous analysis as well as the group sleep timing analyses. Similarly to Olds et al. (2011) [19], we found that sleep timing may play an important and separate role in regulating health factors that influence the risk for being overweight. Contrary to some other studies [30], we found that bedtime in particular may play an important role in the health and development of youth [18] by helping to buffer against the risk of overweight/obesity in youth. Therefore, sleep timing, sleep duration, and other sleep factors may be important to consider in future sleep and obesity research and in developing targeted health interventions.

The strong connection between bedtime and sleep duration seen on weeknights and on weekend nights suggests that bedtime seems to be the driving factor for determining how much sleep youth get on a given night. Because of school times and work times being closer in timing for the majority of people, youth and adults often have less wiggle room to stretch out their morning wake times because they have to get up and get ready to be somewhere at a time that is relatively similar across most groups. This means that people who choose to go to bed late at night, whether based on time use and habits, personal preferences, sociocultural norms, or circadian rhythms, may be likely to get less sleep overall because they have to wake up to be somewhere. The correlations were slightly lower on weekends, suggesting that there may be more flexibility to sleep in on weekends when they do not have to be at school or work at a certain time. These findings highlight the importance of considering bedtime as an important part of establishing healthy sleep habits and patterns [15,18,19,20].

Although the current study does not directly address the mechanisms that cause sleep to impact overweight risk, it does add to the literature on the association between sleep duration, sleep timing, and overweight by adding these findings with an understudied population. The mechanisms that link sleep and overweight risk are not fully delineated in the literature, but researchers have stated that insufficient sleep and late sleep are related to a host of factors, including changes in social and personal time use, as well as changes in circadian rhythms, eating behaviors and hormones [16,17,18,21,22,39,40].

Some study limitations should be acknowledged. First, the sleep measures used in this study were self-reported. Self-reported sleep measures are subject to biases such as social desirability bias and recall bias. To account for these, youth reported basic sleep information—their usual bedtimes and wake times on school days and on weekend days. Then we used their usual bedtimes and wake times to calculate their average sleep duration. Other studies have used self-report for sleep variables in children and youth from ages eight and above and have found that they demonstrate adequate reliability and validity, especially when reporting more basic sleep information (such as bedtimes and wake times) [41,42]. Second, the data used in this study came from a cross-sectional design, which limits causal inferences because we cannot establish temporal precedence between sleep variables and risk for overweight. Third, the study examines a very specific population, Midwestern Latine youth, which may limit generalizability to other populations. While this provides valuable insights for this specific population, these findings may differ for other groups or for youth in other regions. Therefore, more work is needed in different populations to see how it relates to the findings in the current study. Finally, the use of dichotomized variables and smaller groups in the replication of the previous study [19] reduced the statistical power in the analyses. To address this limitation, we also included analyses where we used the direct continuous variables in separate analyses to examine the effects of sleep duration and sleep timing on overweight risk.

Though there are limitations, this study also has many strengths. First, the study used an objective measure of BMI. These objective measures are less likely to be biased and are more reliable, increasing the overall strength of the study. Second, this study was designed to focus on differences within one culture instead of comparing across cultures, which sometimes leads to views that one culture is healthier or better than another culture. These paucity or scarcity views generally pit one group against another and promote inequities, leading to more harm than good in some cases. Instead, we focused on how behaviors within this unique and underrepresented group led to different outcomes so that we can promote positive routines such as earlier bedtimes and stay relevant to this specific population while being as culturally sensitive as possible. This included building initial rapport with the community of interest through establishing relationships with people that families in the communities already knew and trusted, traveling to where these families lived (often up to an hour away), meeting the families in their homes, speaking their language, and providing questionnaires in both English and Spanish. Finally, we followed a similar design and analytical approach to an existing study [19], with an effort to replicate results from their study in a new population and to build on the literature in this area. This is a particularly difficult group to recruit because the majority of the families live in rural areas in the Midwest that are not close to any universities, and some members of this population are reticent to participate in research and hesitant to allow strangers in their homes. This could be due to fears surrounding immigration policies or cultural differences in understanding what researchers are trying to learn. Additionally, language differences and lack of rapport were also barriers to accessing this community, and we worked very hard over several years to build the rapport necessary, and to recruit enough bilingual research assistants, for us to collect this data from the homes of the participants. These efforts all represent strengths of the inclusion of this population and the study design that was necessary to access this data.

Future research of these variables among this population is warranted, particularly with the inclusion of objective sleep measures such as sleep actigraphy devices. Recent studies have shown that most of the current research in these groups continue to utilize primarily self-report and parent-report data [30,41,42]. Though self-report sleep variables can be considered adequate, objective measures alone or paired with self-report sleep variables may strengthen future studies and increase our understanding of the mechanisms of sleep and their connections to health [41,42]. This may be particularly important to use in studies with Latine youth and other minority populations, where less is known and there is more need for building a literature around sleep functions and mechanisms [43]. Other sociocultural factors may also play important roles in establishing sleep habits and norms in Latine youth, including factors such as living in multigenerational households, community expectations surrounding sleep hygiene and sleep needs, household size and density, neighborhood factors, and more [30,43]. Researchers may also want to consider other factors in minority youth populations, such as temperature and noise, which have been shown to influence sleep quality even when one is going to bed early enough or staying in bed long enough to get healthy sleep [43].

Employing longitudinal, experimental designs would allow researchers to explore potential long-term trends and causal implications in the relationship between sleep and obesity. Additional research should examine the potential biological and sociocultural factors contributing to higher rates of obesity and to delayed sleep timing, specifically in Midwestern Latine youth. Larger studies with samples of children from similar populations and from diverse racial and ethnic backgrounds could allow researchers to examine both within-group differences and across-group differences in culturally sensitive ways. This may help speak to racial and ethnic similarities and differences in positive ways and build our knowledge base about the sociocultural determinants of sleep factors such as timing and duration, and whether these findings are similar for other minority groups [29,30]. These findings can inform policy development aimed at reducing health disparities and disproportionate rates of obesity and overweight for minority groups, as well as helping parents and educators understand the impact sleeping behaviors have on the health of their children [42].

Finally, as a field there is still a need for future research to establish the mechanisms that link sleep duration, sleep timing, and other factors to overweight risk and other health outcomes. Some studies have suggested that sleep timing can influence circadian rhythm misalignment that then increases hunger hormones (ghrelin) and decreases sensitivity to satiety hormones (leptin), leading to excessive eating and craving calorie dense and unhealthy foods [38,39]. Additionally, these metabolic changes can trigger a change in the HPA axis in the body that regulates stress hormones (cortisol), as well as decreasing the body’s insulin sensitivity [39], both of which have been associated with increased risk for being overweight and unhealthy. In this view, a lack of sleep would increase the body’s perception of stress and raise the levels of cortisol, then impacting hunger hormones and blood sugar levels in unhealthy ways. This could harm metabolism and lead to increased calorie consumption, putting the body in a calorie surplus and leading to unhealthy weight gain. In theory, that weight gain could then impact the quality of sleep and ability to get adequate physical activity (without a lot of discomfort and/or pain) and lead to increased weight gain, becoming a cyclical problem that worsens across the lifespan.

## 5. Conclusions

We found that suboptimal sleep duration, later bedtimes, and late sleep timing patterns were associated with risk for being overweight in Midwestern Latine youth. Late-to-bed and late-to-rise youth had the highest overweight risk. Although these findings come from a cross-sectional study that uses self-report data, and focuses only on one specific population, it supports the need for further studies on this topic. It is especially important to continue this research on sleep factors and overweight risk in similar populations and other populations where less focus has been given. If more studies establish similar patterns between sleep duration, bedtimes, and risk for being overweight, it will give stronger support for public health programs and future interventions to include sleep interventions as part of a holistic approach to lowering the risk for being overweight. Sleep represents a modifiable risk behavior that can either be detrimental to health or improve health, and there remains to be a need for examining these connections in minority populations, where less is known about both sleep and obesity. We encourage other researchers, practitioners, and policy makers to consider the benefits of addressing multiple factors surrounding sleep routines, focusing on increasing knowledge regarding the role of sleep timing in addition to sleep duration, especially in underserved minoritized populations.

## Figures and Tables

**Table 1 children-13-00032-t001:** Means and Correlations for BMI Percentile, Sleep, and Demographic Variables for Midwestern Latine Youth.

Variable	Mean	*SD*	1	2	3	4	5	6	7	8	9	10	11
1. Child BMI Percentile	81.57	21.68	—										
2. Sleep duration—WD	9 h 12	0 h 56	−0.14	—									
3. Sleep duration—WE	10 h 13	1 h 32	−0.11	**0.33 ****	—								
4. Bedtime—WD	9:36 PM	0 h 47	0.10	**−0.77 ****	**−0.25 ****	—							
5. Wake time—WD	6:47 AM	0 h 36	−0.09	**0.55 ****	**0.18 ***	0.11	—						
6. Bedtime—WE	10:58 PM	1 h 19	**0.18 ^†^**	**−0.49 ****	**−0.63 ****	**0.50 ****	−0.11	—					
7. Wake time—WE	9:11 AM	1 h 15	0.05	−0.11	**0.57 ****	**0.22 ***	0.11	**0.28 ****	—				
8. Child Age (years)	11.53	0.69	0.09	**−0.22 ***	0.05	**0.23 ***	−0.05	−0.02	0.04	—			
9. Child Gender ^a^	1.59	0.49	**−0.31 ****	−0.05	**0.18** ^†^	−0.02	−0.09	−0.03	**0.19 ***	0.16	—		
10. Maternal Education ^b^	2.48	1.48	−0.08	0.06	−0.04	**−0.17** ^†^	−0.15	−0.11	**−0.17** ^†^	−0.02	0.02	—	
11. Household Income	$27,500	$15,000	0.05	0.04	−0.09	−0.09	−0.08	−0.04	−0.15	−0.08	−0.02	**0.31 ****	—

Note: Sample size *N* = 119. WD = Weekday, WE = Weekend, Boldface indicates statistical significance ** *p* < 0.01, * *p* < 0.05, ^†^ *p* < 0.10. ^a^ Child gender was coded as boys = 1, girls = 2. Overall, the sample consisted of 70 girls (59%), and 49 boys (41%). ^b^ The average level of maternal education was some high school. Of the 119 mothers, only 36 had completed high school or more.

**Table 2 children-13-00032-t002:** Associations between Sleep Categories, Sleep Duration, Stress, and Overweight.

**Weekday Sleep Patterns**							
**Variable**	**EE** **(*n* = 29)**	**EL** **(*n* = 33)**	**LE** **(*n* = 27)**	**LL** **(*n* = 30)**	**All (*n* = 119)**	** *F* **	** *p* ** **-Value**
Sleep duration (hours/day)	9 h 17 ±43 min	10 h 07 ±37 min	8 h 13 ±36 min	9 h 00 ±38 min	9 h 12 ±57 min	**44.27**	**<0.001**
Bedtime	8:56 PM ±34 min	9:05 PM ±24 min	10:12 PM ±29 min	10:16 PM ±34 min	9:35 PM ±47 min	**58.24**	**<0.001**
Wake time	6:13 AM ±29 min	7:12 AM ±23 min	6:25 AM ±21 min	7:15 AM ±19 min	6:47 AM ±36 min	**56.39**	**<0.001**
Overweight prevalence	51.7% ±50.9%	51.5% ±50.8%	59.3% ±50.1%	80.0% ±40.7%	60.5% ±49.1%	***X*^2^ = 6.84**	**0.077**
**Weekend Sleep Patterns**							
**Variable**	**EE** **(*n* = 34)**	**EL** **(*n* = 21)**	**LE** **(*n* = 31)**	**LL** **(*n* = 33)**	**All (*n* = 119)**	** *F* **	** *p* ** **-Value**
Sleep duration (hours/day)	10 h 20 ±62 min	12 h 10 ±42 min	8 h 42 ±70 min	10 h 15 ±71 min	10 h 13 ±93 min	**44.77**	**<0.001**
Bedtime	9:48 PM ±37 min	10:08 PM ±25 min	11:38 PM ±51 min	12:06 AM ±75 min	10:58 PM ±79 min	**50.42**	**<0.001**
Wake time	8:08 AM ±29 min	10:18 AM ±40 min	8:20 AM ±39 min	10:21 AM ±42 min	9:11 AM ±75 min	**90.88**	**<0.001**
Overweight prevalence	50.0% ±50.8%	38.1% ±50.0%	74.2% ±44.5%	72.7% ±45.2%	60.5% ±49.1%	***X*^2^ = 10.48**	**0.015**

Note: Sample size *N* = 119. Boldface indicates statistical significance. EE = Early-bed/Early-rise; EL = Early-bed/Late-rise; LE = Late-bed/Early-rise; LL = Late-bed/Late-rise. These unadjusted ANOVA comparisons were performed across the groups to compare direct differences between the groups on these key variables as a reference for readers.

**Table 3 children-13-00032-t003:** Associations between Sleep Duration, Sleep Timing, and Overweight Prevalence in Midwestern Latine Youth.

Analysis	Prevalence of Overweight % (95% CI)	OR (95% CI)	OR (95% CI)
Child Overweight ^c^ By Sleep Duration		Unadjusted ^a^	Multivariate adjusted ^b^
Weeknight Sleep		**0.67 (0.44–1.02)** ^†^	**0.53 (0.31–0.91) ***
Weekend Night Sleep		**0.69 (0.52–0.90) ****	**0.67 (0.49–0.91) ***
Child Overweight By Sleep Timing		Unadjusted ^a^	Multivariate adjusted ^b^
Weekday bedtime		**1.86 (1.10–3.13) ***	**2.35 (1.28–4.31) ****
Weekday wake time		1.01 (0.55–1.87)	0.93 (0.45–1.89)
Weekend bedtime		**1.45 (1.05–2.00) ***	**1.66 (1.15–2.40) ****
Weekend wake time		0.85 (0.63–1.15)	0.93 (0.65–1.32)
Child Overweight By Sleep Category		Unadjusted ^a^	Multivariate adjusted ^b^
Weekday Early-Bed	**51.6% (38.8–64.4) ^d^**	1.00 (reference)	1.00 (reference)
Weekday Late-Bed	**70.2% (57.9–82.4)**	**2.21 (1.04–4.69) ***	**3.12 (1.02–8.08) ***
** *t* ** ** = 2.10 ***			
Weekday Early-Rise	55.4% (43.8–68.8)	1.00 (reference)	1.00 (reference)
Weekday Late-Rise	65.1% (62.0–76.8)	1.50 (0.72–3.15)	1.44 (0.61–3.40)
*t =* −1.08			
Weekend Early-Bed	**45.5% (31.9–59.0)**	1.00 (reference)	1.00 (reference)
Weekend Late-Bed	**73.4% (62.3–84.6)**	**3.32 (1.54–7.15) ****	**3.49 (1.43–8.54) ****
** *t =* ** ** 3.19 ****			
Weekend Early-Rise	61.5% (49.4–73.7)	1.00 (reference)	1.00 (reference)
Weekend Late-Rise	59.3% (45.7–72.8)	0.91 (0.44–1.90)	1.10 (0.46–2.60)
*t =* 0.25			

Note: Sample size *N* = 119 youth. Boldface indicates statistical significance (^†^ *p* < 0.10; * *p* < 0.05, ** *p* < 0.01). ^a^ Unadjusted logistic regression analyses. ^b^ Adjusted logistic regression analyses for child race (sample is all Latine youth), child sex, child age, education level of primary caregiver, family income, and maternal BMI. ^c^ Overweight defined as BMI percentile scores ≥ 85th percentile. ^d^ Unweighted between group independent *t*-tests.

**Table 4 children-13-00032-t004:** Associations between Sleep Timing and Weight Status in Midwestern Latine Youth.

Analysis	Prevalence of Overweight ^a^ (95% CI)	Odds Ratio (95% CI)	Odds Ratio (95% CI)
Child Overweight By Weekday Sleep Timing		Unadjusted ^b^	Multivariate adjusted ^c^
Early/Early	51.72%	1.00 (reference)	1.00 (reference)
Early/Late	51.52%	0.99 (0.37–2.69)	0.82 (0.25–2.74)
Late/Early	59.26%	1.36 (0.47–3.91)	1.72 (0.49–6.10)
**Late/Late**	**80.00%**	**3.73 (1.18–11.83) ***	**4.94 (1.26–19.30) ***
Child Overweight By Weekend Sleep Timing		Unadjusted ^b^	Multivariate adjusted ^c^
Early/Early	50.00%	1.00 (reference)	1.00 (reference)
Early/Late	38.10%	0.62 (0.20–1.86)	0.92 (0.24–3.59)
**Late/Early**	**74.19%**	**2.88 (1.01–8.20) ***	**3.45 (1.01–11.79) ***
**Late/Late**	**72.73%**	**2.67 (0.96–7.39)** ^†^	**3.32 (1.00–11.01) ***

Note: Sample size *N* = 119 youth. Boldface indicates statistical significance (^†^ *p* < 0.10; * *p* < 0.05). ^a^ Overweight defined as BMI percentile scores ≥ 85th percentile. ^b^ Unadjusted logistic regression analyses. ^c^ Adjusted logistic regression analyses for child race (sample is all Latine youth), child sex, child age, education level of primary caregiver, family income, and maternal BMI.

## Data Availability

The raw data supporting the conclusions of this article will be made available by the authors on request. The data are not publicly available due to privacy and ethical reasons.

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
