# Peer review of "Does Bedtime Really Matter? Examining How Sleep Timing Relates to Sleep Duration and Overweight Status in Midwestern Latine Youth"

_children, 2025, doi:10.3390/children13010032_

Round 1
Reviewer 1 Report
Comments and Suggestions for Authors
Thank you for the opportunity to review this manuscript. The study addresses a relevant topic—sleep timing and overweight risk in Latine youth—and has potential value. However, several aspects of the manuscript require clarification and substantial revision to improve methodological transparency, analytical rigor, and alignment with MDPI format standards. Please consider the following point-by-point comments:
Lines 46–47 (Introduction)
The statement “Childhood obesity is a growing epidemic in the U.S. and presents a substantial public health concern”requires a supporting citation.
Lines 52–53 (Introduction)
The definition of obesity provided from Kopelman (2000)—“BMI of 30 kg/m²”—is an adult-specific threshold and is not applicable to children or adolescents. In pediatric populations, obesity is determined using age- and sex-adjusted BMI percentiles (e.g., ≥95th percentile according to CDC or WHO standards). Including an adult definition may cause conceptual inconsistency in a manuscript focused on youth. I recommend removing this sentence or replacing it with a definition aligned with pediatric criteria.
Lines 62–97 (Introduction)
The introduction does not sufficiently articulate the conceptual mechanisms that may explain why sleep timing—beyond sleep duration—relates to obesity risk. Although several studies are cited, the manuscript lacks a clear theoretical framework addressing circadian misalignment, chronobiological regulation, or behavioral pathways that could link later bedtimes to increased adiposity in youth.
Additionally, while the authors state that Latine youth, experience disproportionate obesity risk, the introduction does not clearly justify why immigrant Latine children represent a distinct population requiring focused examination. The manuscript should elaborate on the specific sociocultural, environmental, or structural factors that may influence sleep behaviors and obesity pathways in this group. Without this justification, the motivation for limiting the study to Latine immigrant youth is not fully supported.
Lines 107–134 (Materials and Methods)
The Methods section does not explicitly describe the study design. For clarity and adherence to reporting standards, the authors should clearly specify the study design at the beginning of this section.
Lines 107–122 (Materials and Methods)
The manuscript does not specify the sampling strategy used to recruit participants, nor does it provide any sample size justification or power calculation. For methodological transparency, the authors should clearly report the sampling approach and indicate whether any a priori sample size estimation was conducted.
Lines 108–122 (Materials and Methods)
Although inclusion criteria are described, the manuscript does not report any exclusion criteria. To improve methodological transparency and reproducibility, the authors should clearly specify whether exclusion criteria were defined and, if so, what conditions or circumstances led to participant exclusion.
Lines 123–133 (Materials and Methods)
The Procedure section provides a general description of the home visits but lacks several methodological details necessary for reproducibility. Specifically, the manuscript does not clarify the standardized protocol used across research assistants, the timing and conditions under which anthropometric measurements were taken, or how incomplete or inconsistent survey responses were handled. Additionally, it is not stated whether all participants followed an identical protocol.
Lines 135–146 (Materials and Methods)
The Measures section provides a basic description of BMI and sleep variables but lacks several important methodological details. First, sleep timing and duration rely solely on child self-report, yet no information is provided regarding the reliability or validity of these measures in this age group. Second, the anthropometric protocol is insufficiently described: although CDC procedures are mentioned, the manuscript does not specify the equipment used, calibration procedures, measurement conditions, or whether duplicate measurements were taken. Additionally, the manuscript references waist circumference in the abstract but does not explain how this variable was measured. Finally, no psychometric information is provided for the questionnaires administered in English and Spanish.
Lines 148–157 (Statistical Analysis)
The detailed description of how the four sleep-timing categories (EE, EL, LE, LL) were constructed appears in the Statistical Analysis section; however, this information pertains to variable operationalization rather than to statistical procedures. For clarity and proper structuring of the Methods, the construction of these categories should be described earlier in the Measures section (or in a dedicated subsection), and the Statistical Analysis section should focus strictly on the analytical models used.
Lines 147–161 (Statistical Analysis)
The Statistical Analysis section lacks several key methodological details. Although odds ratios are reported in the results, the manuscript does not explicitly state that logistic regression models were used, nor does it describe model diagnostics, handling of missing data, or assessment of assumptions. Additionally, the use of ANOVA to examine the relationship between sleep timing and BMI is not clearly justified. ANOVA is a group-comparison test, yet none of the study’s objectives or hypotheses involve comparing mean BMI values across groups; instead, all stated hypotheses focus on associations with overweight status, which is a dichotomous outcome. As such, ANOVA does not align with the analytic framework required to test the hypotheses presented. The rationale for dichotomizing bedtime and wake-time based on medians should also be justified, as this approach may reduce statistical power and obscure meaningful variability.
Lines 162–219 (Results)
The Results section presents relevant findings; however, several issues limit clarity and methodological rigor. First, the heavy reliance on median-based dichotomization of continuous variables (bedtime and wake-time) reduces statistical power and may obscure important dose–response patterns. Second, the inclusion of ANOVA group comparisons is not aligned with the study’s predefined hypotheses, which focus exclusively on overweight status as a dichotomous outcome. Third, the coding of overweight as a continuous variable in Table 1 is conceptually inappropriate and should be revised. Additionally, the results do not address potential multicollinearity among sleep variables, despite strong correlations presented in Table 1.
Lines 225–274 (Discussion)
The Discussion summarizes the main findings but lacks depth in interpretation and critical analysis. Much of the text reiterates results rather than explaining the underlying mechanisms linking sleep timing to overweight risk. A more robust integration of circadian, behavioral, and metabolic pathways would strengthen the scientific contribution. Additionally, the Discussion does not address key methodological limitations—such as the median-based dichotomization of sleep variables, potential multicollinearity, and the exclusive use of child self-report for sleep measures. The section also underdevelops the rationale for focusing on Latine immigrant youth, missing an opportunity to contextualize findings within relevant sociocultural determinants.
Lines 275–282 (Conclusions)
The Conclusion summarizes the primary findings but could be strengthened by adopting a more cautious tone consistent with the study’s cross-sectional design and self-reported measures of sleep. Several statements imply intervention-oriented or causal interpretations that are not fully supported by the data. Additionally, the Conclusion does not explicitly tie back to the study’s hypotheses or highlight the unique contribution of examining sleep timing in rural Latine youth. Incorporating these elements and acknowledging key limitations would enhance the clarity and scientific rigor of this section.
Author Response
Responses to Reviewer 1
Thank you for the opportunity to review this manuscript. The study addresses a relevant topic—sleep timing and overweight risk in Latine youth—and has potential value. However, several aspects of the manuscript require clarification and substantial revision to improve methodological transparency, analytical rigor, and alignment with MDPI format standards. Please consider the following point-by-point comments:
Comment 1: Lines 46–47 (Introduction)
The statement “Childhood obesity is a growing epidemic in the U.S. and presents a substantial public health concern”requires a supporting citation.
Response 1: Thank you for catching this. We added our first references there to lead readers to examples of why obesity is a public health concern.
Comment 2: Lines 52–53 (Introduction)
The definition of obesity provided from Kopelman (2000)—“BMI of 30 kg/m²”—is an adult-specific threshold and is not applicable to children or adolescents. In pediatric populations, obesity is determined using age- and sex-adjusted BMI percentiles (e.g., ≥95th percentile according to CDC or WHO standards). Including an adult definition may cause conceptual inconsistency in a manuscript focused on youth. I recommend removing this sentence or replacing it with a definition aligned with pediatric criteria.
Response 2: Thank you! We agree that only listing adult criteria here would be confusing or misleading. We added some sentences distinguishing how to calculate BMI percentile in children and youth (page 2, lines 60-68)
Comment 3: Lines 62–97 (Introduction)
The introduction does not sufficiently articulate the conceptual mechanisms that may explain why sleep timing—beyond sleep duration—relates to obesity risk. Although several studies are cited, the manuscript lacks a clear theoretical framework addressing circadian misalignment, chronobiological regulation, or behavioral pathways that could link later bedtimes to increased adiposity in youth.
Additionally, while the authors state that Latine youth, experience disproportionate obesity risk, the introduction does not clearly justify why immigrant Latine children represent a distinct population requiring focused examination. The manuscript should elaborate on the specific sociocultural, environmental, or structural factors that may influence sleep behaviors and obesity pathways in this group. Without this justification, the motivation for limiting the study to Latine immigrant youth is not fully supported.
Response 3: Thank you for pointing out these omissions. We agree that the paper needed more attention to each of these details. We have added about 8 new references that directly address each of these issues and go into more depth throughout the paper about the mechanisms linking sleep and obesity, as well as the importance of this group in our study. We hope these additions adequately respond to these needs.
Comment 4: Lines 107–134 (Materials and Methods)
The Methods section does not explicitly describe the study design. For clarity and adherence to reporting standards, the authors should clearly specify the study design at the beginning of this section.
Response 4: Thank you for noting this need. We added more information about the study throughout the method section and clarified the study design.
Comment 5: Lines 107–122 (Materials and Methods)
The manuscript does not specify the sampling strategy used to recruit participants, nor does it provide any sample size justification or power calculation. For methodological transparency, the authors should clearly report the sampling approach and indicate whether any a priori sample size estimation was conducted.
Response 5: Thank you for this suggestion. We did use a power analysis prior to data collection in this main study and have now included this information in the participants section of the paper. We also gave more details about the recruitment plan and why we selected the number of participants and our recruiting strategy in the participants section.
Comment 6: Lines 108–122 (Materials and Methods)
Although inclusion criteria are described, the manuscript does not report any exclusion criteria. To improve methodological transparency and reproducibility, the authors should clearly specify whether exclusion criteria were defined and, if so, what conditions or circumstances led to participant exclusion.
Response 6: We did not have any exclusion criteria. This was such a difficult and time-consuming sample to acquire data from that we tried to be as inclusive as possible. We note this with a new section in the participants section, as well as adding a highlight of this difficulty of recruitment to the paragraph on strengths in the discussion. Thank you for pointing this out. We hope these additions add to the quality and strength of the paper.
Comment 7: Lines 123–133 (Materials and Methods)
The Procedure section provides a general description of the home visits but lacks several methodological details necessary for reproducibility. Specifically, the manuscript does not clarify the standardized protocol used across research assistants, the timing and conditions under which anthropometric measurements were taken, or how incomplete or inconsistent survey responses were handled. Additionally, it is not stated whether all participants followed an identical protocol.
Response 7: Thank you for this suggestion. We added more details surrounding the recruitment process, the training of the RAs, the procedure for the home visits, and the measurements in the participants, procedure, and measures sections. We hope that this makes the paper more transparent and easier to replicate.
Comment 8: Lines 135–146 (Materials and Methods)
The Measures section provides a basic description of BMI and sleep variables but lacks several important methodological details. First, sleep timing and duration rely solely on child self-report, yet no information is provided regarding the reliability or validity of these measures in this age group. Second, the anthropometric protocol is insufficiently described: although CDC procedures are mentioned, the manuscript does not specify the equipment used, calibration procedures, measurement conditions, or whether duplicate measurements were taken. Additionally, the manuscript references waist circumference in the abstract but does not explain how this variable was measured. Finally, no psychometric information is provided for the questionnaires administered in English and Spanish.
Response 8: Thank you for these suggestions. We took out the waist circumference comment in the abstract. Sorry for that confusion. We added information in the paper that talks about the validity and reliability of youth-reported sleep variables (in the discussion). We added a lot more detail about the anthropometric measurements and tools we used (in the procedure section). We used study measures that have been validated in Spanish and English in the larger study that this data were taken from, but for the measures used in this study we had the measures translated to Spanish and then back-translated to ensure clarity. It was really only the demographic questions and the sleep questions for the youth. As noted in the paper, almost all of the youth answered their questionnaires in English anyway, but we made sure that the Spanish versions were clear and lined up with the English versions prior to data collection. We hope that these additions and clarifications are helpful to the reader, and we feel that it improves the quality and clarity of the paper. Thank you for noting these excellent suggestions.
Comment 9: Lines 148–157 (Statistical Analysis)
The detailed description of how the four sleep-timing categories (EE, EL, LE, LL) were constructed appears in the Statistical Analysis section; however, this information pertains to variable operationalization rather than to statistical procedures. For clarity and proper structuring of the Methods, the construction of these categories should be described earlier in the Measures section (or in a dedicated subsection), and the Statistical Analysis section should focus strictly on the analytical models used.
Response 9: Thank you. We agree that this needed more clarity and a subsection of its own. We moved it into the measures section and provided more details about how these groups were created. The Statistical Analysis section now focuses on the statistics that we ran for each analysis.
Comment 10: Lines 147–161 (Statistical Analysis)
The Statistical Analysis section lacks several key methodological details. Although odds ratios are reported in the results, the manuscript does not explicitly state that logistic regression models were used, nor does it describe model diagnostics, handling of missing data, or assessment of assumptions. Additionally, the use of ANOVA to examine the relationship between sleep timing and BMI is not clearly justified. ANOVA is a group-comparison test, yet none of the study’s objectives or hypotheses involve comparing mean BMI values across groups; instead, all stated hypotheses focus on associations with overweight status, which is a dichotomous outcome. As such, ANOVA does not align with the analytic framework required to test the hypotheses presented. The rationale for dichotomizing bedtime and wake-time based on medians should also be justified, as this approach may reduce statistical power and obscure meaningful variability.
Response 10: Thank you for noting the need for more clarity and detail here. We added that we used logistic regression models and explained how we ran these analyses. Next, we added a note about missing data. We only had very minor missing data values on two of the control variables (none of the main sleep or BMI variables). We used imputation to assign the mean value to both of those variables (for any missing values) and described this in the first paragraph of the plan for analysis. For the assumptions, we ensured that the variables did not have any excessive skewness or kurtosis (we added this sentence in the paper as well). We added a paragraph in the text, and a note under Table 4, about why we included the ANOVA analyses. It was not to change the study hypotheses, but rather to provide a visual representation of how the groups compare across main study variables. This is to help researchers and readers understand more basic information about how the groups lined up without having to include all of that information in the paper itself. If the reviewers and editor feel that it is distracting in any way and would be better to remove it, then we are willing to take it out. We hope that with more clarification for why it is there that it provides more meaning and value. Finally, we added more information about how and why we dichotomized the variables to create the group comparisons. We did this to replicate work done in the Olds et al. (2011) paper, but also included continuous variables in different analyses so that we could show both ways of testing these connections between sleep duration, sleep timing, and overweight risk. We felt that the replication of the Olds paper and the addition of the continuous variables of sleep allowed us to look at both views on sleep duration and sleep timing. This added additional strength to the study by showing that the categorical sleep timing group analyses lined up with the sleep timing directly seen in the continuous variable analysis, and that this showed that it was bedtimes driving this relationship (rather than wake times). We added more information about this important distinction in the discussion (due to your comment), and feel that this improved the paper clarity and value. Thank you for pointing out the need for making these adjustments. We felt that these suggestions alone greatly improved the paper quality.
Comment 11: Lines 162–219 (Results)
The Results section presents relevant findings; however, several issues limit clarity and methodological rigor. First, the heavy reliance on median-based dichotomization of continuous variables (bedtime and wake-time) reduces statistical power and may obscure important dose–response patterns. Second, the inclusion of ANOVA group comparisons is not aligned with the study’s predefined hypotheses, which focus exclusively on overweight status as a dichotomous outcome. Third, the coding of overweight as a continuous variable in Table 1 is conceptually inappropriate and should be revised. Additionally, the results do not address potential multicollinearity among sleep variables, despite strong correlations presented in Table 1.
Response 11: Thank you for these suggestions. We added some discussion about the use of dichotomous variables to use for the replication focus of the study, while also including the continuous variables to use as ways to look at overall sleep duration and sleep timing variables. We feel that it strengthens the results by showing both views of sleep timing, and acknowledge that the statistical power can be reduced by breaking the participants into groups in a sample that does not include thousands of participants (like the Olds study). This was such a time and resource intensive sample to collect data from that it would not be feasible in this study to have 2,200 participants. But we made use of our resources as much as possible and wanted to show both kinds of analysis for clarity. We made a note (at the end of the plan for analysis section) about the ANOVA analyses being used more for a visual representation to describe the sleep timing groups, and not being primarily to adjust or deviate from the original hypotheses. We also changed the correlation table to include BMI percentile (continuous variable) instead of overweight status (dichotomous variable). Finally, the multicollinearity issue in Table 1 shows that sleep duration is strongly related to bedtime. This is an important factor and we add a note about this in the discussion section. This is primarily due to the wake times being so similar across the sample, especially on weekdays. Because of school starting at a similar time for most of the youth, the bedtime really drives the total sleep duration because they generally wake up at approximately similar times. There is much less variation in wake time compared to bedtime. Thank you for these suggestions. We agree that these additions and clarifications improve the paper significantly.
Comment 12: Lines 225–274 (Discussion)
The Discussion summarizes the main findings but lacks depth in interpretation and critical analysis. Much of the text reiterates results rather than explaining the underlying mechanisms linking sleep timing to overweight risk. A more robust integration of circadian, behavioral, and metabolic pathways would strengthen the scientific contribution. Additionally, the Discussion does not address key methodological limitations—such as the median-based dichotomization of sleep variables, potential multicollinearity, and the exclusive use of child self-report for sleep measures. The section also underdevelops the rationale for focusing on Latine immigrant youth, missing an opportunity to contextualize findings within relevant sociocultural determinants.
Response 12: Thank you for this suggestion. We added several paragraphs and context about these issues in the discussion now and feel that it is much stronger with these additions and changes. We also added to the limitations, strengths, and future discussion section and added context about the rationale for studying Latine youth.
Comment 13: Lines 275–282 (Conclusions)
The Conclusion summarizes the primary findings but could be strengthened by adopting a more cautious tone consistent with the study’s cross-sectional design and self-reported measures of sleep. Several statements imply intervention-oriented or causal interpretations that are not fully supported by the data. Additionally, the Conclusion does not explicitly tie back to the study’s hypotheses or highlight the unique contribution of examining sleep timing in rural Latine youth. Incorporating these elements and acknowledging key limitations would enhance the clarity and scientific rigor of this section.
Response 13: Thank you for this excellent suggestion. We agree that the conclusion needed these elements to be stated clearly. We have added the more cautious tone throughout the conclusion and highlighted the limitations of using cross-sectional data and self-report data. We also tied the results back to the hypotheses as suggested. We feel that these changed improved the final portion of the paper and thank you for these suggestions.
Reviewer 2 Report
Comments and Suggestions for Authors
Thank you for the opportunity to review this timely and well-written manuscript examining associations between sleep timing, sleep duration, and overweight status among Midwestern Latine youth. The study provides valuable insight into an understudied population, and the use of objective BMI measurement and clear presentation of results—especially the tables—is a strength of the work. The manuscript would benefit from several clarifications and refinements to strengthen its scientific contribution and interpretive accuracy.
The introduction offers a broad overview of the sleep–obesity literature but could be more focused in justifying the unique contribution of examining sleep timing versus duration. The rationale for emphasizing rural Midwestern Latine youth would be enhanced by integrating more literature on sociocultural or environmental determinants of sleep within minoritized communities. Some references are repeated, and streamlining the narrative would improve clarity. In the Methods section, additional justification is needed for the analytic decisions—particularly the use of median splits to create sleep timing categories, which reduces variability and may obscure meaningful differences. These categories follow Olds et al., but acknowledging their limitations and providing a clearer rationale for covariate selection would help situate the analysis conceptually. Because sleep measures are self-reported, it would also be helpful to note this limitation earlier in the Methods rather than only in the Discussion.
The Results are clearly presented and easy to interpret. However, a few elements merit clarification—for example, the unusually long sleep duration in the weekend EL group (Table 4) and whether this reflects true behavior or potential reporting artifacts. When discussing odds ratios, it may strengthen the narrative to consistently restate the reference categories. The Discussion effectively summarizes the findings, but some interpretations extend beyond what cross-sectional data allow. Claims suggesting that sleep timing may be “as important or more important” than sleep duration should be softened, as this comparison was not formally tested. More careful avoidance of causal language would improve the rigor of the interpretation. At the same time, the manuscript could expand on the sociocultural factors that may influence sleep behaviors in this specific community, which would increase the relevance and contextual depth of the findings.
Overall, this is a promising and well-presented manuscript that addresses important questions in pediatric sleep and obesity research. With clearer justification of methodological decisions, tighter framing of the introduction and discussion, and more cautious interpretation of the findings, the study will make a strong contribution to the literature.
Comments on the Quality of English LanguageThe manuscript is generally well written and easy to follow. The English is clear throughout, although several sentences could be streamlined to reduce repetition—particularly in the introduction and discussion. Minor grammatical edits and improved transitions between ideas would further enhance readability. Overall, the quality of writing is solid and requires only light revision.
Author Response
Responses to Reviewer 2
Comment 1: Thank you for the opportunity to review this timely and well-written manuscript examining associations between sleep timing, sleep duration, and overweight status among Midwestern Latine youth. The study provides valuable insight into an understudied population, and the use of objective BMI measurement and clear presentation of results—especially the tables—is a strength of the work. The manuscript would benefit from several clarifications and refinements to strengthen its scientific contribution and interpretive accuracy.
Response 1: Thank you for your thoughtful review and suggestions. We hope these revisions make the paper stronger now and help it to be easier for readers to understand.
Comment 2: The introduction offers a broad overview of the sleep–obesity literature but could be more focused in justifying the unique contribution of examining sleep timing versus duration. The rationale for emphasizing rural Midwestern Latine youth would be enhanced by integrating more literature on sociocultural or environmental determinants of sleep within minoritized communities. Some references are repeated, and streamlining the narrative would improve clarity. In the Methods section, additional justification is needed for the analytic decisions—particularly the use of median splits to create sleep timing categories, which reduces variability and may obscure meaningful differences. These categories follow Olds et al., but acknowledging their limitations and providing a clearer rationale for covariate selection would help situate the analysis conceptually. Because sleep measures are self-reported, it would also be helpful to note this limitation earlier in the Methods rather than only in the Discussion.
Response 2: Thank you for these excellent suggestions and questions. We added a section on theory to the introduction and expanded the context of why we need to keep examining sleep timing vs sleep duration by adding in a new reference (Sunwoo et al, #9) – that found different results from Olds et al. We tied this into our use of this particular sample of Midwestern Latine youth in the following paragraphs on page 3. We took out a few references that were redundant and added some new references that hopefully relate more clearly throughout the paper (such as the Mitchell et al, Sunwoo et al, Jones et al 2014, Bronfenbrenner 1979, Harrison et al., Blanco et al.). For the methods section, we added a note that the median splits were not only to replicate Olds et al., but also to give us the most evenly split groups to use in our analyses. This was important because our sample was not using 2200 people like they did. So we needed to keep groups as relatively similar in size as we could. We also added a limitation note about this in the statistical analysis section (2.4). We added more information about the need for addressing the knowledge gap in Latine youth and sleep (see page 3 in the introduction, where we also added the Blanco reference). We also added more information about the need to include more sociocultural determinants and objective studies in minority samples in the future directions section of the discussion.
Comment 3: The Results are clearly presented and easy to interpret. However, a few elements merit clarification—for example, the unusually long sleep duration in the weekend EL group (Table 4) and whether this reflects true behavior or potential reporting artifacts. When discussing odds ratios, it may strengthen the narrative to consistently restate the reference categories. The Discussion effectively summarizes the findings, but some interpretations extend beyond what cross-sectional data allow. Claims suggesting that sleep timing may be “as important or more important” than sleep duration should be softened, as this comparison was not formally tested. More careful avoidance of causal language would improve the rigor of the interpretation. At the same time, the manuscript could expand on the sociocultural factors that may influence sleep behaviors in this specific community, which would increase the relevance and contextual depth of the findings.
Response 3: Thank you for noting these concerns and asking for clarification. The reason the EL group has such long sleep is that these are the kids who go to bed in the earlier group (median split) and then they wake up in the later group (median split). They end up going to bed around 9:00pm on average, but then sleep until just after 7:00am on average. Their bedtime is similar to the EE group, but they sleep later and wake up the same time as the LL group. This is similar, in effect, to the LE group getting an hour shorter of sleep (they go to bed late but wake up early). To help with the reference categories, we added the full names in the Table 3, clarify them in the note in Table 4 and now restate the reference categories in the results section (as you noted), wherever we talk about the differences of the late to bed groups. We agree that the discussion needs to be softened. We have addressed that by changing our wording about our results to be more cautious and correlational, and we also added more information about sociocultural factors in our explanations, and the need to address more sociocultural factors about sleep in future studies. This includes the Nguyen-Rodriguez reference and future directions discussion about the needs to look beyond the current sleep variables and consider sociocultural factors that have not been addressed adequately in the current research.
Comment 4: Overall, this is a promising and well-presented manuscript that addresses important questions in pediatric sleep and obesity research. With clearer justification of methodological decisions, tighter framing of the introduction and discussion, and more cautious interpretation of the findings, the study will make a strong contribution to the literature.
Response 4: Thank you for this comment. We agree that it has the potential to contribute to the previous literature. We went back to the introduction and added a paragraph on the theoretical support for this study (page 4, lines 126-137). We also worked to address tightening the introduction and discussion with our wording, including adding more caution in the interpretation of the findings (especially based on the sample size and sample characteristics).
Comment 5: Comments on the Quality of English Language
The manuscript is generally well written and easy to follow. The English is clear throughout, although several sentences could be streamlined to reduce repetition—particularly in the introduction and discussion. Minor grammatical edits and improved transitions between ideas would further enhance readability. Overall, the quality of writing is solid and requires only light revision.
Response 5: Thank you for your suggestions and comments. We have gone through the manuscript and tried to reduce repetition throughout (especially in the introduction and conclusion). We hope these changes help the paper to be more concise and easier to read.
Reviewer 3 Report
Comments and Suggestions for Authors
The presentation of the results in Table 3 is unclear, and the structure does not allow readers to follow the logic or interpret the findings reliably. At this stage, I am unable to fully understand the analytical framework or evaluate the scientific value of the manuscript. I recommend that the authors reorganize Table 3 with clearer variable definitions, consistent formatting, and an explanation of how each result should be interpreted.
Author Response
Responses to Reviewer 3
Comment 1: The presentation of the results in Table 3 is unclear, and the structure does not allow readers to follow the logic or interpret the findings reliably. At this stage, I am unable to fully understand the analytical framework or evaluate the scientific value of the manuscript. I recommend that the authors reorganize Table 3 with clearer variable definitions, consistent formatting, and an explanation of how each result should be interpreted.
Response 1: Thank you for this suggestion. The general format is typical of an Odds Ratio table, but in the previous draft we submitted, we forgot to include the overweight percentages of each of the 4 sleep groups. We hope this is helpful. The logistic regression compares the other 3 groups to the reference group (Early to bed and Early to rise). The numbers in the parentheses are the confidence intervals for each group. If those numbers don’t cross 1.00 within that range, then they are significant. If they are higher than 1 then it means that that group has that percentage of an increased risk (odds ratio) compared to the reference group for being overweight. If it is less than 1.00, then it means a decreased risk for being overweight. We present the unadjusted numbers first, and then the adjusted analyses (with the control variables listed below the table). We put the significant results in bold to help them stand out more. The top half of the table is the odds ratio analysis for sleep on weeknights, and the bottom half of the table is for weekend nights. We explain the results of this key analysis in the results, where it talks about Hypotheses 3.
Round 2
Reviewer 1 Report
Comments and Suggestions for Authors
Thank you for submitting the revised version of your manuscript. The paper has improved substantially in clarity, methodological transparency, and theoretical grounding. Most of the major concerns raised in the previous review have been carefully addressed, particularly those related to the description of procedures, justification of the study population, and clarification of the analytical approach. The manuscript is now considerably stronger and closer to being suitable for publication.
That said, a small number of important issues remain that should be addressed to ensure full conceptual coherence, methodological rigor, and alignment with reporting standards.
- Although the data originate from a larger prospective project, the analyses presented in this manuscript rely exclusively on baseline (Time 1) data. As such, the study should be explicitly described as a cross-sectional observational design. Currently, the Methods section may still lead readers to interpret the design as prospective.
- The Introduction includes an adult-based definition of obesity (BMI ≥ 30 kg/m², cited from Kopelman), followed by a correct explanation of pediatric BMI percentiles. Retaining an adult threshold in a manuscript focused exclusively on children and adolescents may cause conceptual confusion. The authors should either remove this adult-based definition or state explicitly that it applies “in adults.” The pediatric percentile-based definition is sufficient and more appropriate for the population examined in this study.
- The Discussion appropriately notes the strong correlation between bedtime and sleep duration. However, the Statistical Analysis section does not report any formal assessment of multicollinearity or other model diagnostics. Please indicate whether multicollinearity was formally evaluated (e.g., variance inflation factors) and briefly report the results. Additionally, clarifying whether key assumptions of logistic regression (e.g., linearity of the logit for continuous predictors) were assessed would further strengthen the methodological rigor.
Author Response
Responses to Reviewers - Round 2
General Comment: Thank you for submitting the revised version of your manuscript. The paper has improved substantially in clarity, methodological transparency, and theoretical grounding. Most of the major concerns raised in the previous review have been carefully addressed, particularly those related to the description of procedures, justification of the study population, and clarification of the analytical approach. The manuscript is now considerably stronger and closer to being suitable for publication.
General Response: Thank you for your follow-up review and positive feedback.
Comment 1: That said, a small number of important issues remain that should be addressed to ensure full conceptual coherence, methodological rigor, and alignment with reporting standards. Although the data originate from a larger prospective project, the analyses presented in this manuscript rely exclusively on baseline (Time 1) data. As such, the study should be explicitly described as a cross-sectional observational design. Currently, the Methods section may still lead readers to interpret the design as prospective.
Response 1: Thank you for this comment. We agree that the information from other waves could have been confusing to readers. We took out that other information, and now we pose this study as a cross-sectional observational design. We also took out the information about measures that were not used in this study (these changes are located in the participants and procedures sections in the method section).
Comment 2: The Introduction includes an adult-based definition of obesity (BMI ≥ 30 kg/m², cited from Kopelman), followed by a correct explanation of pediatric BMI percentiles. Retaining an adult threshold in a manuscript focused exclusively on children and adolescents may cause conceptual confusion. The authors should either remove this adult-based definition or state explicitly that it applies “in adults.” The pediatric percentile-based definition is sufficient and more appropriate for the population examined in this study.
Response 2: Thank you for pointing out this need for clarification in the introduction. We changed the reference here (from Kopelman to Birken et al., 2017) and the wording. Now we talk more about BMI percentiles in youth and further clarify that these percentiles are needed in pediatric populations.
Comment 3: The Discussion appropriately notes the strong correlation between bedtime and sleep duration. However, the Statistical Analysis section does not report any formal assessment of multicollinearity or other model diagnostics. Please indicate whether multicollinearity was formally evaluated (e.g., variance inflation factors) and briefly report the results. Additionally, clarifying whether key assumptions of logistic regression (e.g., linearity of the logit for continuous predictors) were assessed would further strengthen the methodological rigor.
Response 3: Thank you for pointing out this omission in our previous paper. We did test for multicollinearity initially but did not include it in the paper. We have now included this information in the Statistical Analysis section, and elaborate here as well. When we ran the linear regression for these variables and tested the VIF, our sleep variables (sleep duration and bedtime) on weekdays had VIF scores of 2.37 and 2.47 (respectively), and our weekend sleep duration and bedtime were even lower, with 1.707 and 1.67 respectively. These numbers are below even the typical VIF score concerns that range from 10 (normal concern level) down to 5 (more conservative VIF score concern level) [e.g., Kim, 2019 – PMID:31304696]. Based on those numbers, we decided that it was appropriate to leave both variables in the models. As Vatcheva et al. (2016; PMID:27274911) note, it is important to put the variables into the same regression model and see if the VIF values increase. The numbers above represent the VIF scores with the variables in the same models, so those are the highest VIF scores we saw.
Additionally, we performed the Box-Tidwell test and compared the transformations to ensure that we did meet the assumptions for linearity of the logit in our continuous variables in logistic regression. We added a note about this to the Statistical Analysis section as well. Thank you for this suggestion.
